# CHAT-3D: DATA-EFFICIENTLY TUNING LARGE LANGUAGE MODEL FOR UNIVERSAL DIALOGUE OF 3D SCENES

## ABSTRACT

3D scene understanding has gained significant attention due to its wide range of applications. However, existing methods for 3D scene understanding are limited to specific downstream tasks, which hinders their practicality in real-world applications. This paper presents Chat-3D, which combines the 3D visual perceptual ability of pre-trained 3D representations and the impressive reasoning and conversation capabilities of advanced LLMs to achieve the first universal dialogue systems for 3D scenes. Specifically, we align 3D representations into the feature space of LLMs, thus enabling LLMs to perceive the 3D world. Given the scarcity of 3D scene-text data, we propose a three-stage training strategy to efficiently utilize the available data for better alignment. To enhance the reasoning ability and develop a user-friendly interaction scheme, we further construct a high-quality object-centric 3D instruction dataset and design an associated object-centric prompt. Our experiments show that Chat-3D achieves an impressive ability to comprehend diverse instructions for 3D scenes, engage in intricate spatial reasoning, and incorporate external knowledge into its responses. Chat-3D achieves a 82.2% relative score compared with GPT-4 on the constructed instruction dataset.

## 1 INTRODUCTION

3D vision is an important way for robots to perceive the rich semantic and spatial information of the real world. 3D scene understanding (Azuma et al., 2022; Ma et al., 2022; Chen et al., 2020; Achlioptas et al., 2020; Chen et al., 2021) has garnered increasing attention in recent years, owing to its broad range of applications in human-robot interaction, metaverse, robotics, and embodied intelligence. However, current methods (Wang et al., 2023a;b; Yang et al., 2021; Jiao et al., 2022; Yuan et al., 2022; Parelli et al., 2023) are limited in addressing specific downstream tasks, such as captioning and question answering, while lacking the ability to engage in general dialogue regarding a 3D scene, restricting their practicality in various real-world tasks. A universal dialogue system for 3D scenes is an imperative component of high-level intelligent robots.

The general dialogue system for 3D scenes requires two kinds of abilities: 3D perception and reasoning. Recently, several studies (Yu et al., 2022; Pang et al., 2022; Wang et al., 2021; Zhang et al., 2022; Xue et al., 2023; Liu et al., 2023b) on pre-trained 3D representations shows impressive performance in 3D perception. However, the reasoning ability for the 3D world remains constrained owing to the scarcity of reasoning and describing data for 3D scenes.

Large language models (LLMs) (Chiang et al., 2023; OpenAI, 2023; Touvron et al., 2023; Chowdhery et al., 2022), on the other hand, exhibit remarkable prowess in complex reasoning and open-domain conversations. Moreover, recent methods (Li et al., 2023b; Liu et al., 2023a; Zhao et al., 2023; Zhang et al., 2023a; Zhu et al., 2023) attempt to extend LLMs to image and video fields. These works typically adopt a two-stage training scheme: Firstly, the visual representations are aligned into the word embedding space of LLMs by leveraging large-scale image-text and video-text datasets (Lin et al., 2014; Sharma et al., 2018; Changpinyo et al., 2021; Schuhmann et al., 2021; 2022; Bain et al., 2021; Miech et al., 2019; Xu et al., 2016). Secondly, they enhance the reasoning capabilities of LLMs regarding visual concepts by fine-tuning on the instruction datasets.

Despite the success of image and video understanding fields, introducing LLMs to perceive 3D scenes faces two challenges: 1) Compared to the millions or even billions of image-text and video-text data (Sharma et al., 2018; Changpinyo et al., 2021; Schuhmann et al., 2021; 2022; Bain et al., 2021), the 3D scene-text data (Achlioptas et al., 2020; Chen et al., 2020) is limited. Consequently, in the low-resource scenarios, the commonly used two-stage training scheme in previous multi-modal LLMs is less effective in aligning pre-trained 3D representations to the feature space of LLMs. 2) 3D scenes always encompass a greater number of objects compared to an image or a video clip. Thus, the common questions or instructions in images and videos are more susceptible to ambiguity in 3D scenes. Consider a simple question like "What is in front of this chair?" on a 3D scene that contains multiple chairs. The dialogue model cannot understand which specific chair the user is asking about, and uniquely describing an object (the chair) in question is often difficult and user-unfriendly due to the complex object relations.

In this paper, we propose Chat-3D, the first attempt to extend the reasoning and conversation capabilities of LLMs to 3D scene understanding. We employ a three-stage training scheme to more efficiently utilize the limited data. Specifically, in the first stage, we directly align the features of 3D objects with the word embeddings of their class names. In the second stage, we learn a 3D object relation module via 3D scene-text data to capture semantic information about the whole 3D scene. Finally, in the third stage, we further tune the model with a high-quality instruction dataset. To further enhance the reasoning ability of Chat-3D, we construct the instruction dataset via an object-centric scheme, which means all instructions are related to a specific object. Combining our object-centric prompt, users can effortlessly select the object in the scene they want to engage in a dialogue about, without the need to uniquely describe the specific object in their instructions.

In summary, our contributions can be summarized as follows:

(1) We build the first universal dialogue system for 3D scenes, leveraging the advanced visual perception capabilities of 3D pre-trained models, in conjunction with the powerful reasoning and open-domain conversational abilities of LLMs.

(2) We introduce a new three-stage training scheme for multi-modal LLM, enabling the model to progressively transition from learning individual object attributes to capturing complex spatial object relations. This approach effectively improves the quality of dialogue with limited available data.

(3) We construct a high-quality object-centric 3D instruction dataset including diverse dialogues about object attributes, positions, relationships, functionalities, placement suggestions, and detailed descriptions within 3D scenes. We propose a corresponding object-centric prompt approach to provide a user-friendly interaction method.

(4) Our experiments demonstrate that Chat-3D exhibits remarkable capabilities in universal dialogue and spatial reasoning based on 3D scenes. We also employ quantitative comparison to evaluate the effectiveness of our three-stage training scheme and instruction dataset.

## 2 RELATED WORK

**3D Representation Learning**    3D point cloud is a fundamental visual modality. Recently, numerous attempts are made to learn discriminative and robust representations for point cloud objects. PointBERT (Yu et al., 2022), Point-MAE (Pang et al., 2022), Transformer-OcCo (Wang et al., 2021), and point-m2ae (Zhang et al., 2022) employ self-supervised learning approaches to extract meaningful representations of 3D objects from unlabeled point cloud data. Another series of works aims to extend representation from other modalities to 3D. For instance, ULIP (Xue et al., 2023) and openshape (Liu et al., 2023b) construct (3D-image-text) triplets to align point clouds within the CLIP (Radford et al., 2021; Cherti et al., 2023) representation space, while I2P-MAE (Zhang et al., 2023b) and ACT (Dong et al., 2023) learn 3D representations from image pre-trained models (Dosovitskiy et al., 2020; He et al., 2016). These powerful 3D representations can effectively capture the detailed information of a 3D object. In Chat-3D, we segment the 3D scene into objects and extract features for each object, which yields a set of object features to represent the 3D scene and serves as a prerequisite for an object-centric interactive approach.

**3D-Language Tasks**    The interaction between 3D point clouds and natural language has wild applications and has garnered increasing attention recently. 3D captioning (Chen et al., 2021; 2020;

Achlioptas et al., 2020) focuses on generating descriptions of a specific object in a 3D scene. In 3D visual question answering (Azuma et al., 2022), the model is required to answer questions based on the visual content of the 3D scene, while the more complex 3D situated question answering (Ma et al., 2022) requires the model to understand agent's situation (position, orientation, etc.) in a 3D scene as described by text, reason about the surrounding environment. Different from vision-language tasks (Kazemzadeh et al., 2014; Krishna et al., 2017; Goyal et al., 2017; Antol et al., 2015; Lin et al., 2014; Grauman et al., 2022) and methods (Li et al., 2022; 2023a; 2021; Lin et al., 2022; Deng et al., 2021; Wang et al., 2023c) based on images and videos, these 3D-language tasks and corresponding methods place more emphasis on spatial reasoning and the possible interaction between agents and scenes. Despite the significant progress made in this field, existing methods still focus on improving isolated task-specific models, without exploring a unified dialogue system.

**Multi-modal Large Language Models** Recently, Large Language Models showcase remarkable abilities in complex reasoning and conversational communication with humans. To extend the knowledge, reasoning, and conversation abilities acquired from vast amounts of text data to more modalities, some studies (Li et al., 2023b; Liu et al., 2023a; Zhao et al., 2023; Zhang et al., 2023a; Zhu et al., 2023) attempt to instruction tune LLMs for multimodal learning. Specifically, these works first use the caption learning objective to learn the aligning of visual features with pre-trained LLMs from large-scale vision-language paired data. Then, a high-quality instruction dataset is utilized to further enhance the LLMs' comprehension of the visual world. However, in the 3D-Language field, 3D scene-text pairs are scarce. Thus the indirect aligning method is unreliable and incomplete for 3D representations and pre-trained LLMs. To mitigate this issue, we propose a more data-efficient three-stage tuning scheme that establishes a more direct learning stage for alignment, reduces the annotation requirements, and provides a smooth learning curve.

# 3 METHODS

## 3.1 ARCHITECTURE

Chat-3D aims to create a universal dialogue system for 3D scenes by aligning 3D representations with pre-trained LLM (Touvron et al., 2023). The overall network architecture is illustrated in Figure 1.

For the input 3D scene $S$, we first use a 3D object segmentation model (Jiang et al., 2020; Misra et al., 2021; Qi et al., 2019) or ground truth annotations to segment it into objects. Then, users can select the specific object they want to engage in dialogue. The selected target object is denoted as $o_t$ and other objects in the scene are represented as $O_s = [o_1, o_2, \ldots, o_{n_s}]$, where $n_s$ is the number of other objects in the 3D scene. For each object, we use a pre-trained 3D point encoder $g(\cdot)$ to extract features, Besides, we further incorporate extra object attributes (e.g. color, size, location) into these object features by a projector $f_e(\cdot)$ to enrich semantic information. These 3D object features are projected to the word embedding space of pre-trained LLM via a projector $f_a(\cdot)$. The process of 3D object feature extraction and mapping can be expressed as:

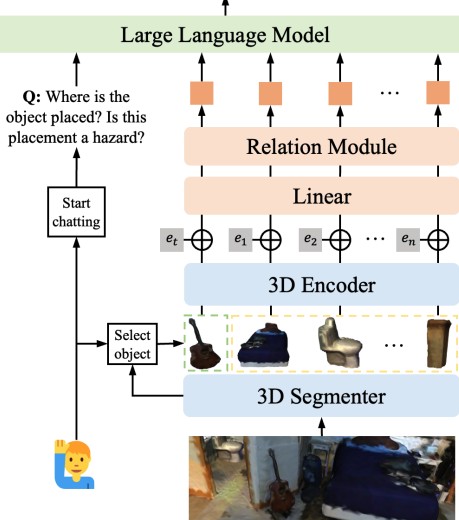

A: The guitar leaned against the wall between the bed and the bathroom. This placement is potentially dangerous, especially when using the toilet at night. A person could trip over a guitar, resulting in a fall or injury.

Figure 1: The overall architecture of Chat-3D.

$$\mathbf{z}_i = f_a(g(o_i) + \mathbf{e}_i), \text{ with } \mathbf{e}_i = f_e([\mathbf{c}_i; \mathbf{s}_i; \mathbf{l}_i]) \quad (1)$$

where $i \in [t, 1, 2, \ldots, n_s]$, and $\mathbf{c}_i, \mathbf{s}_i, \mathbf{l}_i \in \mathbb{R}^3$ respectively represent the RGB value, bounding box size, and location for the $i$-th object. The extracted 3D features of target object and other objects are denoted as $\mathbf{z}_t$ and $\mathbf{Z}_s = [\mathbf{z}_1, \mathbf{z}_2, \ldots, \mathbf{z}_{n_s}]$.

Furthermore, we further introduce a relation module $t(\cdot)$ for capturing complex relations between objects. The features of objects then interact with each other to provide additional object relation information about the scene.

$$[\hat{\mathbf{z}}_t, \hat{\mathbf{z}}_1, \hat{\mathbf{z}}_2, \ldots, \hat{\mathbf{z}}_{n_s}] = r([\mathbf{z}_t, \mathbf{z}_1, \mathbf{z}_2, \ldots, \mathbf{z}_{n_s}]) \qquad (2)$$

The representations of a 3D scene are provided as $\hat{\mathbf{z}}_t \in \mathbb{R}^d$, $[\hat{\mathbf{z}}_1, \hat{\mathbf{z}}_2, \ldots, \hat{\mathbf{z}}_{n_s}] \in \mathbb{R}^{n_s \times d}$, and $d$ is the dimension of hidden states in the pre-trained LLMs.

Lastly, to facilitate user-friendly interaction between our system and users, we design an object-centric prompt as: *###Human: [target] $\hat{\mathbf{z}}_t$ [/target], [scene] $\hat{\mathbf{z}}_1, \hat{\mathbf{z}}_2, \ldots, \hat{\mathbf{z}}_{n_s}$ [/scene], [instruction] ###Assistant:*. Through this prompt, the LLM can comprehend the specific object the user wants to discuss and generate responses based on the 3D visual information and the given instructions.

## 3.2 THREE-STAGE TRAINING

Previous multi-modal LLMs (Li et al., 2023b; Liu et al., 2023a; Zhao et al., 2023; Zhang et al., 2023a; Zhu et al., 2023) primarily follow a two-stage training scheme. In the first stage, LLMs take inputs from visual modality and learn to generate corresponding captions. The large-scale image- and video-text datasets allow comprehensive alignment between visual representations and the word embedding space of LLM. In the second stage, the model is fine-tuned with a high-quality instruction dataset, thereby further enhancing the perceptual and reasoning abilities.

However, in the 3D understanding field, the 3D scene-text data is significantly less than image- or video-text data. For example, the commonly used ScanRefer (Chen et al., 2020) dataset, which provides descriptions for ScanNet (Dai et al., 2017), only contains 36,655 captions for training. In contrast, the datasets used for the first stage training in previous multi-modal LLM methods are million-level or even billion-level, such as CC3M (Sharma et al., 2018), CC12M (Changpinyo et al., 2021), LAION-400M (Schuhmann et al., 2021), LAION-5B (Schuhmann et al., 2022) and WebVid-10M (Bain et al., 2021). Considering the scarcity of 3D scene-text data, we propose a more data-efficient three-stage training approach, which divides the process of aligning 3D features with the pre-trained LLM into two progressive stages: 3D object alignment and 3D scene alignment.

**Stage 1: 3D Object Alignment** The first stage is designed to learn the alignment between the representation of individual 3D objects and pre-trained LLM. Given a 3D object and its annotated category, the 3D object is encoded into a representation $\mathbf{z} \in \mathbb{R}^d$ according to Equation 1. Its category name is encoded into a word embedding $\mathbf{y} \in \mathbb{R}^d$ using the tokenizer of the pre-trained LLM. By maximizing the cosine similarity between the corresponding $\mathbf{z}$ and $\mathbf{y}$, we can learn projectors $f_e(\cdot)$ and $f_a(\cdot)$ that effectively inject the 3D object representations into the word embedding space of LLM.

The advantage of Stage 1 is three-fold: 1) Compared to learning alignment through captioning objective, maximizing the similarity between representations provides a more direct learning objective for alignment, which can achieve more efficient alignment in low-resource scenarios. 2) Stage 1 enables the utilization of 3D point cloud object classification datasets, such as ShapeNet (Chang et al., 2015), ScanObjectNN (Uy et al., 2019), and Objaverse (Deitke et al., 2023), which enhances the model's generalization performance on diverse real-world objects. 3) The introduction of Stage 1 offers a smoother learning curve for comprehending complex 3D scenes. The model progressively transitions from learning individual object attributes to capturing intricate spatial object relations.

**Stage 2: 3D Scene Alignment** After aligning individual 3D object feature with pre-trained LLM, Stage 2 takes a step further by integrating the entire 3D scene into LLM. The training data is sourced from the ScanRefer dataset, which provides annotations for objects in a scene primarily based on their spatial relationships. Considering a 3D scene, which can be segmented into object set $[o_1, o_2, \ldots, o_n]$, we sequentially select each object as target objects and construct the input for LLM according to the methodology discussed in Section 3.1. The instruction in prompts requests the model to generate a brief description of the target object within the 3D scene. The learning objective is to generate a description that aligns with the description provided by the ScanRefer dataset for the target object, and only the two projectors $f_e(\cdot)$, $f_a(\cdot)$ and the relation module $r(\cdot)$ are learnable in this stage.

**Stage 3: Instruction Tuning** For enhancing the reasoning ability about 3D world, we curate a high-quality instruction dataset which comprises rich and detailed instructions. By tuning Chat-3D on this dataset, we further enhance its capability to comprehend diverse instructions, generate imaginative and contextually appropriate responses, engage in intricate spatial reasoning, and effectively incorporate external knowledge into its responses.

# 4 OBJECT-CENTRIC INSTRUCTION DATASET

**Caption of the target object:**
Descriptions: ["There is a single white armchair. placed next to the window of the room.", "The sofa chair is the corner chair. lying parallel to the wall. a small table with the lamp is present beside the chair.", "This is a white sofa chair. it is under a window.", "This is a white armchair. is next to a lamp.", "This is the corner sofa chair. a small table with a lamp can be seen near this chair."]
**Categories and locations of target object and its 10 neighbors:**
Described object: {sofa chair:[-1.31, 3.15, 0.59]}; Neighbor objects: {window:[-1.12, 4.12, 1.59], table:[0.86, 1.61, 0.38], doorframe:[-2.25, 0.67, 1.27], windowsill:[0.88, 3.97, 0.98], windowsill:[-1.32, 3.93, 0.91], sofa chair:[0.98, 3.35, 0.71], window:[1.16, 4.18, 1.73], pillow:[1.35, 0.29, 0.46], table:[-0.15, -2.66, 0.26], tv:[-2.2, -0.55, 1.52]}

Table 1: An example of textualizing an object in a 3D scene

You are an AI 3D visual assistant, and you are seeing an object in a 3D scene. What you see is provided with several sentences, describing the same object you are looking at, and the position of surrounding objects in the 3D scene to represent the content of the 3D scene. Based on these descriptions of this object and the location of surrounding objects in the 3D scene, summary and describe the placement, function of this object, and how a person can access this object in detail as if you are in the 3D scene.
Importantly, do not mention any specific spatial coordinate values. The description should be more than 150 words and less than 200 words.

Table 2: Prompt for descriptive object-centric captions.

Design a conversation between you and a person asking about this object in the 3D scene. The answers should be in a tone that a visual AI assistant is in the 3D scene and answering the question. Ask diverse questions and give corresponding answers.
Include questions asking about the visual content of this object, including the object types, object shape, object attribute, object functions, object locations, relative positions between objects, etc. Only include questions that have definite answers:
(1) Questions whose contents can be confidently observed and answered based on the 3D scene.
(2) Questions whose absence from the 3D scene can be confidently determined.

Table 3: Prompt for object-centric conversations.

The complex object relationships and intricate interactions between agents and scenes impose elevated demands on reasoning capabilities. To enhance the reasoning ability pertaining to 3D world, we construct a high-quality object-centric instruction dataset based on the annotations in ScanRefer. Specifically, we leverage the remarkable reasoning and summarizing capabilities of ChatGPT to automatically generate descriptive and detailed captions as well as diverse conversations centered around specific objects within 3D scenes.

**Object-centric Descriptive Captions**    ScanRefer annotates multiple captions for objects in a 3D scene based on their attributes and spatial relationships. We employ ChatGPT to summarize and rewrite these short captions into imaginative paragraphs. To facilitate ChatGPT's comprehension of the 3D scene, we also textualize the 3D scene as shown in Table 1, providing the categories and XYZ coordinates of the target object and its ten nearest objects in the scene. Furthermore, we propose a prompt to request ChatGPT to focus on perceiving and reasoning about the object relations and agent interactions as exemplified in Table 2. During dataset construction, we initially manually annotated several examples and randomly selected two of them as in-context examples to guide the generation of ChatGPT. One example of the generated descriptive object-centric caption is shown in Table 4.

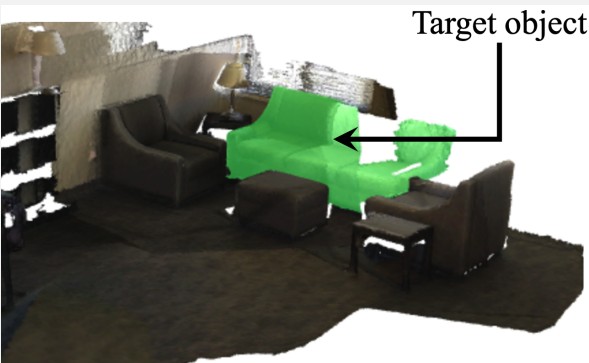

**Detailed description:** The object is a brown couch with three seats, positioned on the left corner of the room. It is next to a gold lamp, which serves as a source of lighting and complements the overall aesthetics of the area. The couch is accompanied by an ottoman placed in front of it, providing a comfortable footrest and enhancing the seating experience. The presence of two armchairs nearby indicates that this space is designed for relaxation and socializing. The grey-colored couch with three cushions exudes a modern and stylish vibe, adding to the room's visual appeal. The strategic placement of the couch and ottoman encourages a cozy and inviting atmosphere, ideal for leisurely activities and gatherings.
This setup creates a comfortable seating arrangement, promoting interaction and conversation among individuals using the couch and armchairs. The nearby window adds natural light to the area, enhancing the ambiance. The presence of an end table with a lamp contributes to the functionality of the space by providing a convenient spot for placing items and lighting.
In summary, the brown couch with its ottoman and neighboring armchairs forms a functional and aesthetically pleasing seating arrangement. The combination of the couch's design, the presence of the ottoman, and the proximity to the end table with a lamp ensures a comfortable and enjoyable experience for individuals using this part of the room.

Table 4: Example of descriptive object-centric caption.

**Object-centric Conversations**    To enhance the capability of handling diverse instructions and general conversations, we further require ChatGPT to autonomously generate multi-turn dialogues in a self-questioning and self-answering format based on the brief captions of the target object and the textualized 3D scene.

## 5    EXPERIMENTS

### 5.1    IMPLEMENTATION DETAILS

During the training phase, we directly use ground truth annotations (point cloud and extra attributes) of each object in the 3D scene for better training quality. We employ the pre-trained Point-Bind(Guo) model with Point-BERT(Yu et al., 2022) architecture as $g(\cdot)$ to extract features for each object. Meanwhile, we use a linear layer as $f_e(\cdot)$ to incorporate extra attributes (such as color, size, and

location) into the extracted features. Then, a two-layer MLP serves as $f_a(\cdot)$ to map these 3D object features to the word embedding space of the pre-trained LLM, and the relation module $r(\cdot)$ is implemented using a one-layer vanilla transformer encoder. It is worth mentioning that the relation module is zero-initialized, thereby preserving the information learned in Stage 1 when Stage 2 begins. The chosen LLM for our experiment is a Vicuna 7B model(Chiang et al., 2023), which is fine-tuned from the LLaMA base model(Touvron et al., 2023).

## 5.2 EVALUATION ON OBJECT-CENTRIC DATASET

**Relative Score rated by GPT-4**  In order to quantitatively evaluate the universal dialogue ability of Chat-3D and analyze the effect of the three-stage training scheme and our instruction dataset, we adopt GPT-4 (OpenAI, 2023) to measure the quality of our Chat-3D's generated responses following LLaVA (Liu et al., 2023a) and miniGPT4 (Zhu et al., 2023). Specifically, we randomly select 30 scenes from the ScanRefer validation set and randomly choose one object as the target object for each scene. We employ the instruction dataset construction methodology described in Section 4 and Chat-3D respectively to generate responses under the same scene and instruction inputs. After that, we input the textualized scene, instructions, and the two kinds of generated responses into GPT-4 and request GPT-4 to provide an overall score on a scale of 1 to 10 for each response based on its helpfulness, relevance, accuracy, and level of detail. A higher score indicates a higher quality of response.

In Table 5, we study the effectiveness of the instruction dataset and compare the Chat-3D trained via our three-stage training scheme and the two-stage training method used in previous methods (Li et al., 2023b; Liu et al., 2023a; Zhao et al., 2023; Zhang et al., 2023a; Zhu et al., 2023). First, our three-stage training scheme significantly outperforms the previous two-stage method by 6.8 points, demonstrating the data efficiency of our three-stage training method in the low-resource setting. Second, by comparing different combination settings of the instruction dataset, we observe that incorporating conversation data leads to a higher improvement in conversation tests, while integrating detailed caption data enhances performance in detailed caption tests. By utilizing all the data together, our model demonstrates proficiency in both conversation and detailed caption tasks, ultimately achieving the highest overall score.

**Caption Score**  In Table 6, we evaluate the caption metrics BLEU, METEOR, and ROUGE-L under various conditions. This evaluation covers the entire dataset, including all conversations and detailed captions, offering a more comprehensive understanding of the significance of different components. It is clear from the table that excluding either conversation data or detailed caption data leads to a significant decrease in performance. Without stage three, performance deteriorates even more dramatically. These results highlight the effectiveness of our proposed three-stage training architecture.

## 5.3 EVALUATION ON SCANQA

In order to evaluate on the ScanQA dataset, we finetune the pretrained Chat-3D to fit the answer format of this dataset.

**Baselines**  We include representative baseline models on the benchmark. Specifically, **ScanQA** (Azuma et al., 2022) utilizes VoteNet to generate object proposals and then integrates them

| Training | Training Data | | Evaluate Set | | Overall |
|----------|---------------|---------------|--------------|------------------|---------|
| scheme | Conversation | Detailed Caption | Conversation | Detailed Caption | |
| Three-Stage | ✓ | ✓ | **88.2** | **76.2** | **82.2** |
| Two-Stage | ✓ | ✓ | 84.8 | 65.9 | 75.4 |
| Three-Stage | ✓ | ✗ | 85.7 | 53.9 | 69.8 |
| Three-Stage | ✗ | ✓ | 85.1 | 69.2 | 77.2 |
| Three-Stage | ✗ | ✗ | 56.8 | 55.0 | 55.9 |

Table 5: Relative scores on the object-centric dataset (rated by GPT-4).

| Training Scheme | Training Data | | BLEU-1 | BLEU-2 | BLEU-3 | BLEU-4 | METEOR | ROUGE-L |
| | Conv. | Detail. | | | | | | |
|---|---|---|---|---|---|---|---|---|
| Three-Stage | ✓ | ✓ | **38.19** | **22.71** | **13.80** | **8.81** | **19.83** | **35.41** |
| Two-Stage | ✓ | ✓ | 36.54 | 21.31 | 12.69 | 7.95 | 19.59 | 34.95 |
| Three-Stage | ✓ | ✗ | 10.63 | 6.79 | 4.51 | 3.12 | 9.84 | 34.57 |
| Three-Stage | ✗ | ✓ | 18.28 | 10.54 | 6.00 | 3.54 | 18.12 | 15.92 |
| Three-Stage | ✗ | ✗ | 1.31 | 0.76 | 0.39 | 0.21 | 3.89 | 17.71 |

Table 6: Caption scores on the object-centric dataset.

| Method | BLEU-1 | BLEU-2 | BLEU-3 | BLEU-4 | METEOR | ROUGE-L | CIDEr |
|---|---|---|---|---|---|---|---|
| VoteNet+MCAN | 28.0 | 16.7 | 10.8 | 6.2 | 11.4 | 29.8 | 54.7 |
| ScanRefer+MCAN | 26.9 | 16.6 | 11.6 | 7.9 | 11.5 | 30.0 | 55.4 |
| ScanQA | 30.2 | 20.4 | 15.1 | 10.1 | 13.1 | 33.3 | 64.9 |
| LLaVA (zero-shot) | 7.1 | 2.6 | 0.9 | 0.3 | 10.5 | 12.3 | 5.7 |
| 3D-LLM (flamingo) | 30.3 | 17.8 | 12.0 | 7.2 | 12.2 | 32.3 | 59.2 |
| 3D-LLM (BLIP2-flant5) | 39.3 | 25.2 | 18.4 | 12.0 | 14.5 | 35.7 | 69.4 |
| Chat-3D | 29.1 | 16.3 | 10.1 | 6.4 | 11.9 | 28.5 | 53.2 |

Table 7: Evaluation results on ScanQA validation set.

with language embeddings. **ScanRefer+MCAN** (Chen et al., 2020) and **VoteNet+MCAN** (Ding et al., 2019) detect 3D objects and incorporate them into a standard VQA model known as MCAN (Yu et al., 2019). **LLaVA** (Liu et al., 2023a) proposes the visual instruction tuning method, which establishes a connection between a vision encoder and LLM to enable general-purpose visual and language understanding. **3D-LLM** (Hong et al., 2023) relies on 2D Vision-Language Models (VLMs) as their backbone, such as flamingo and BLIP-2. It extracts meaningful 3D features from rendered multi-view images, which serve as the input for the VLM.

**Analysis**   Essentially, Chat-3D achieves competitive results compared to fully supervised methods such as ScanQA. The most intriguing aspect is the performance gap between 3D-LLM and Chat-3D. When changing the backbone from Flamingo to BLIP2-flant5, 3D-LLM experiences a significant performance boost across all metrics. This suggests that 3D-LLM heavily relies on the robust 2D VLM, which is pretrained on billion-level data. In contrast, Chat-3D solely utilizes 3D data for pre-training and fine-tuning, which is based on a much smaller data set. Nevertheless, it still manages to achieve competitive results compared to 3D-LLM (Flamingo), highlighting the effectiveness of our training architecture.

## 5.4   QUALITATIVE COMPARISONS & ANALYSIS

In section, we provide visualization examples of conversations about 3D scenes with Chat-3D. From these cases, we mainly study the perception, reasoning, and dialogue capabilities of Chat-3D. Besides, we further compare Chat-3D with 2D multimodal LLM methods such as MiniGPT-4 (Zhu et al., 2023), LLaVA (Liu et al., 2023a), and mPLUG-owl (Ye et al., 2023) to demonstrate the advantages and necessity of developing a specific multi-modal LLM for 3D scenes.

**Perception, Reasoning and Dialogue**   We provide several examples of conversations with Chat-3D in figs. 2 and 3 to 7, covering various commonly-seen object types (*e.g.*, table, chair, and bed). In fig. 2(a), Chat-3D shows strong perception capabilities by accurately counting objects, recognizing shapes, and precisely localizing them within the 3D space. In fig. 4, Chat-3D demonstrates impressive reasoning capabilities by deducing the cabinet's purpose and evaluating its practicality based on its placement and spatial relationships with surrounding objects. Guided by the object-centric prompt outlined in Section 3.1, Chat-3D adeptly directs its attention to the specific target object indicated by the user. This enables Chat-3D to maintain focus on the intended subject without being diverted by other similar objects present in the scene. Moreover, the conversational exchanges consistently demonstrate the high-quality dialogue delivered by Chat-3D.

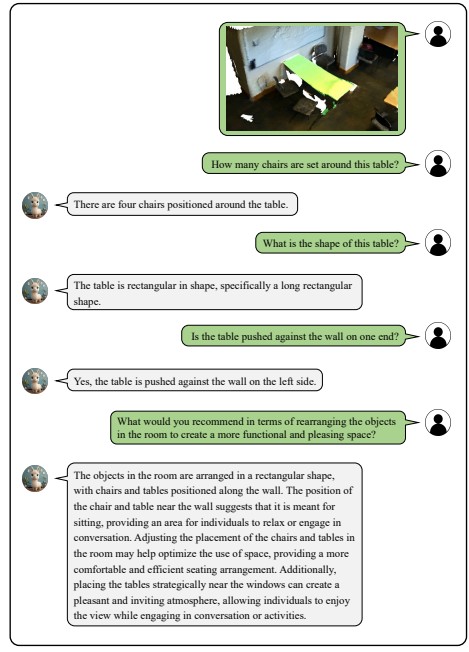

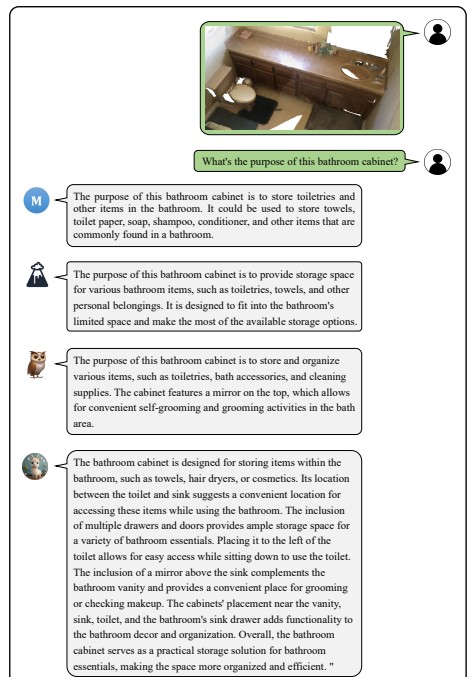

(a) Example 1 of Chat-3D conversation.

(b) Example 1 of comparison between Chat-3D and 2D Multi-modal LLMs.

Figure 2: Qualitative examples of Chat-3D. [1]

**Comparisons with 2D Multi-modal LLMs**   We compare Chat-3D with MiniGPT-4 (Zhu et al., 2023), LLaVA (Liu et al., 2023a), and mPLUG-owl (Ye et al., 2023) in figs. 2 and 8 to 10. As illustrated in fig. 2(b), Chat-3D demonstrates its capability to provide a clear and meticulous analysis of the given question. In Example 2, depicted in fig. 8, we assess the model's spatial perception ability in determining whether both monitors are of identical size. Chat-3D showcases accurate identification, while the other 2D models provide incorrect answers due to their limitations in grasping depth and perspective relationships within the 2D image. In Example 3, presented in fig. 9, the constraints of 2D models become more apparent as they struggle to accurately identify the spatial relationships between the target object and its surrounding objects. Compared to 2D models, Chat-3D's analytical prowess shines brightly due to its remarkable aptitude for perceiving and comprehending concepts within the 3D space.

## 6   CONCLUSION

In this paper, we build the first universal dialogue system for 3D scenes, leveraging the advanced visual perception capabilities of 3D pre-trained models, in conjunction with the powerful reasoning and open-domain conversational abilities of LLMs. To overcome the challenge of limited 3D data availability, we introduce a three-stage training scheme for multi-modal LLMs to progressively transition from learning individual object attributes to capturing complex spatial object relations. Furthermore, we construct a high-quality object-centric 3D instruction dataset and propose a corresponding object-centric prompt approach to facilitate a user-friendly interaction method. Experimental results demonstrate that Chat-3D showcases remarkable capabilities in universal dialogue, spatial reasoning, and the enhancement of external knowledge based on 3D scenes.

---

[1]More examples can be found in the appendix.

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

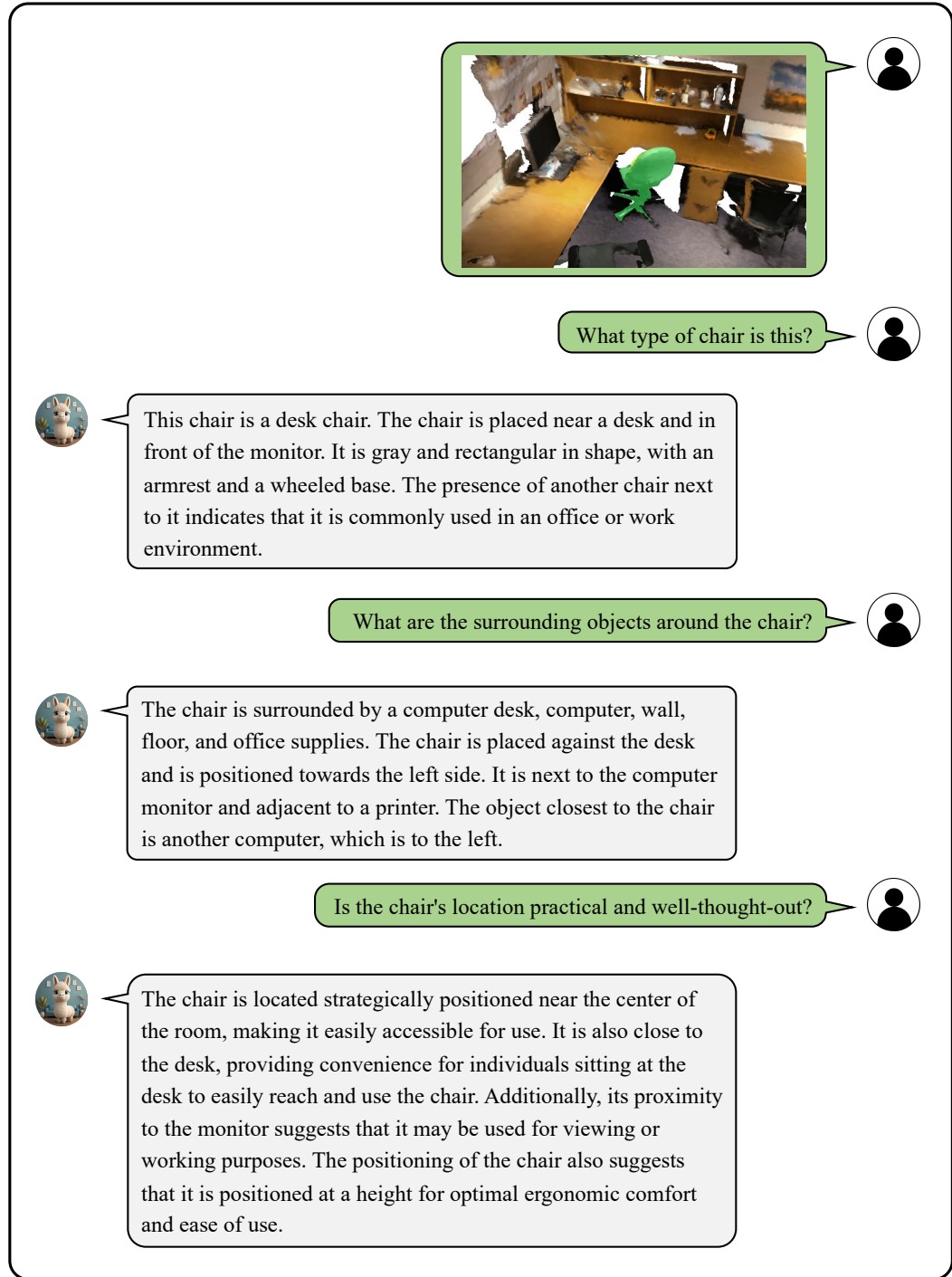

Figure 3: Example 2 of Chat-3D conversation.

# A   APPENDIX

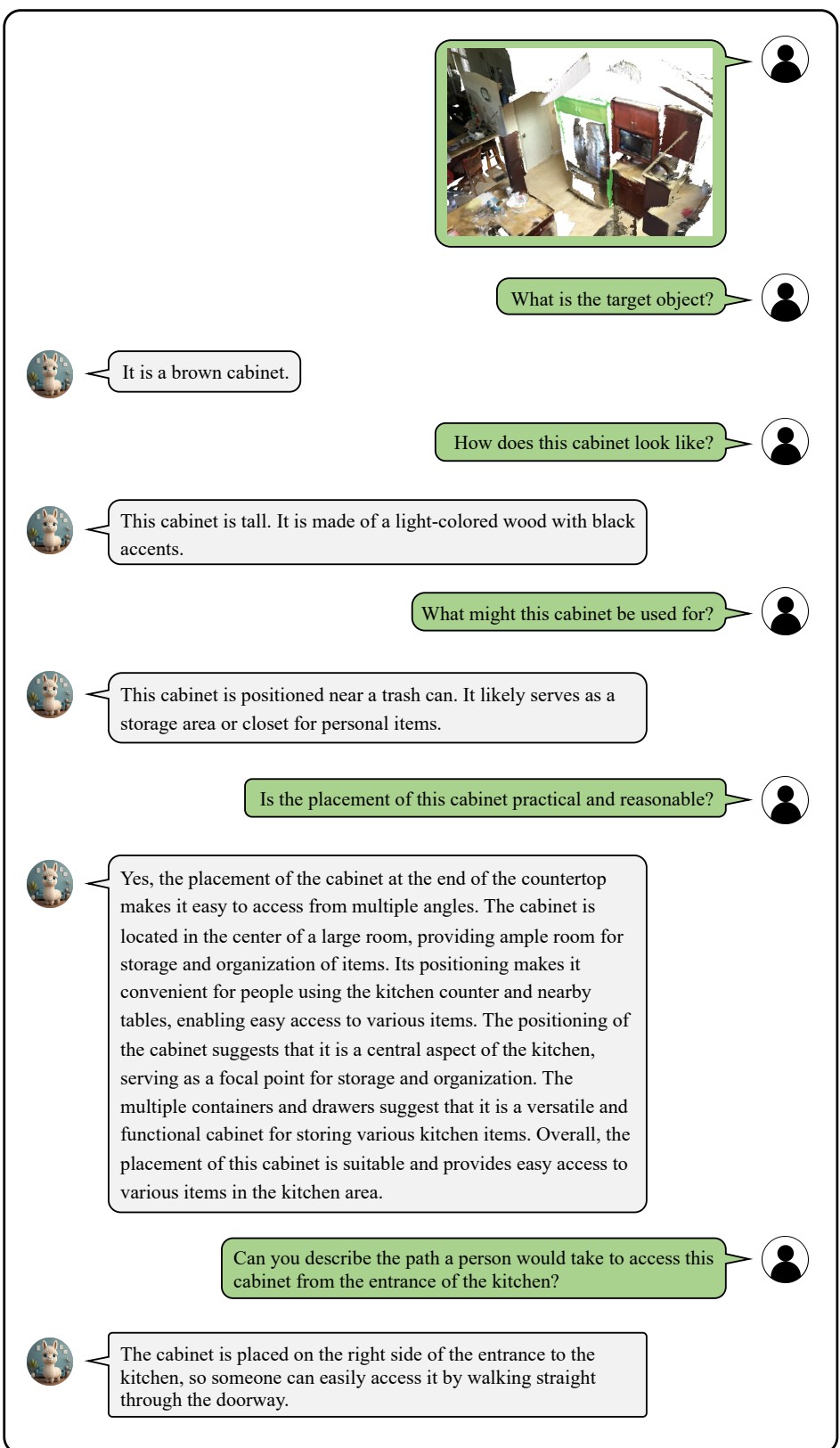

Figure 4: Example 3 of Chat-3D conversation.

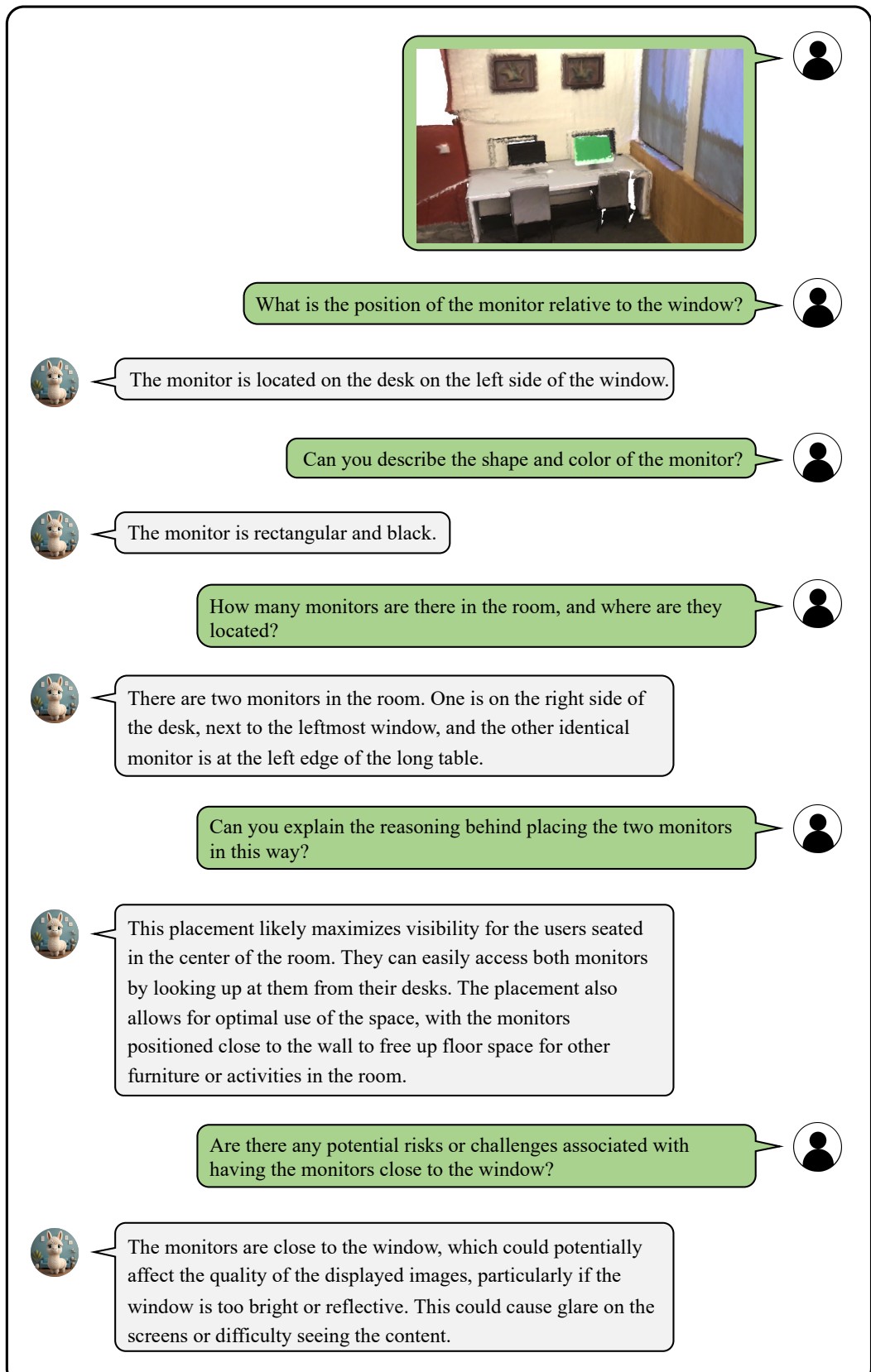

Figure 5: Example 4 of Chat-3D conversation.

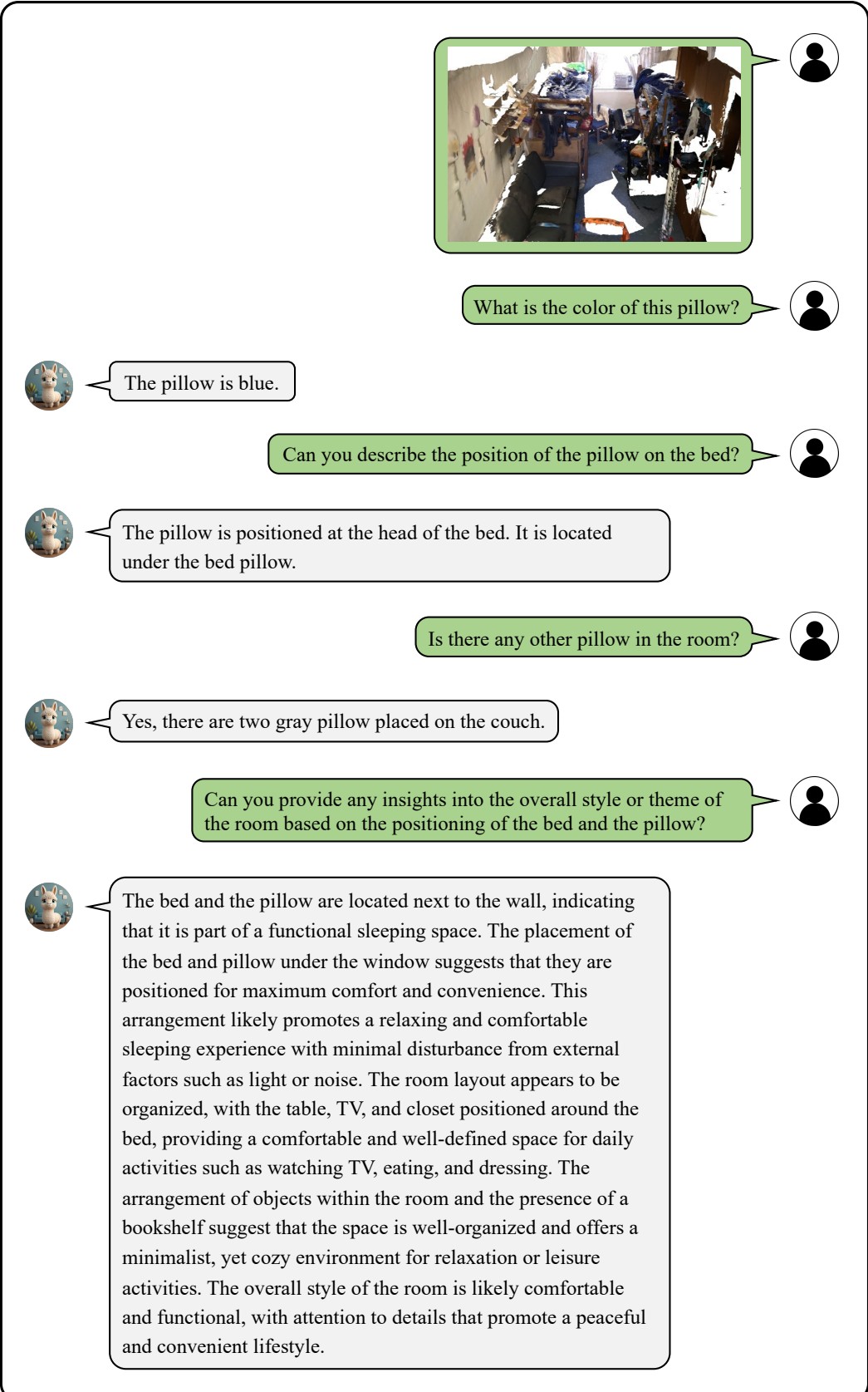

Figure 6: Example 5 of Chat-3D conversation.

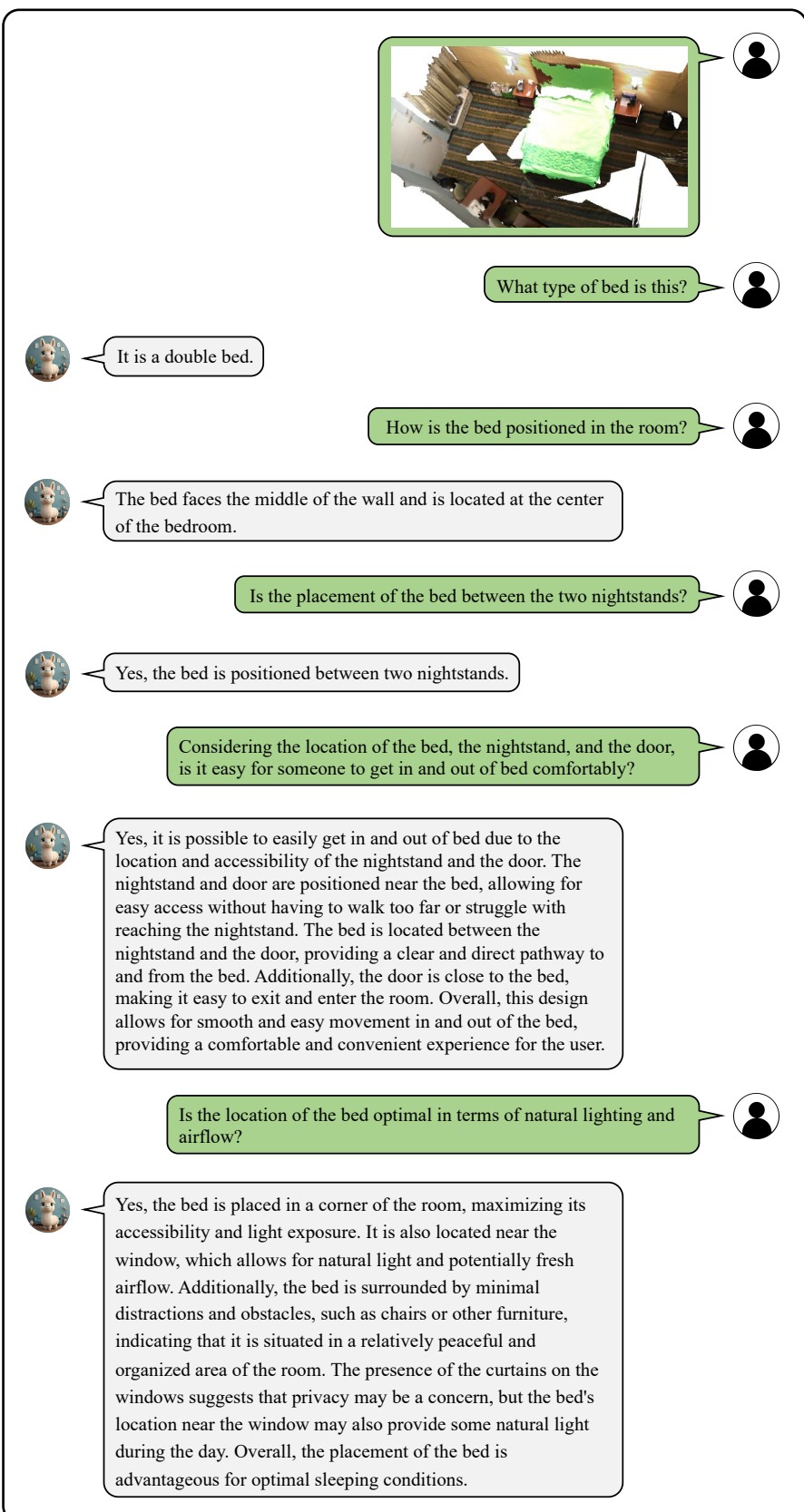

Figure 7: Example 6 of Chat-3D conversation.

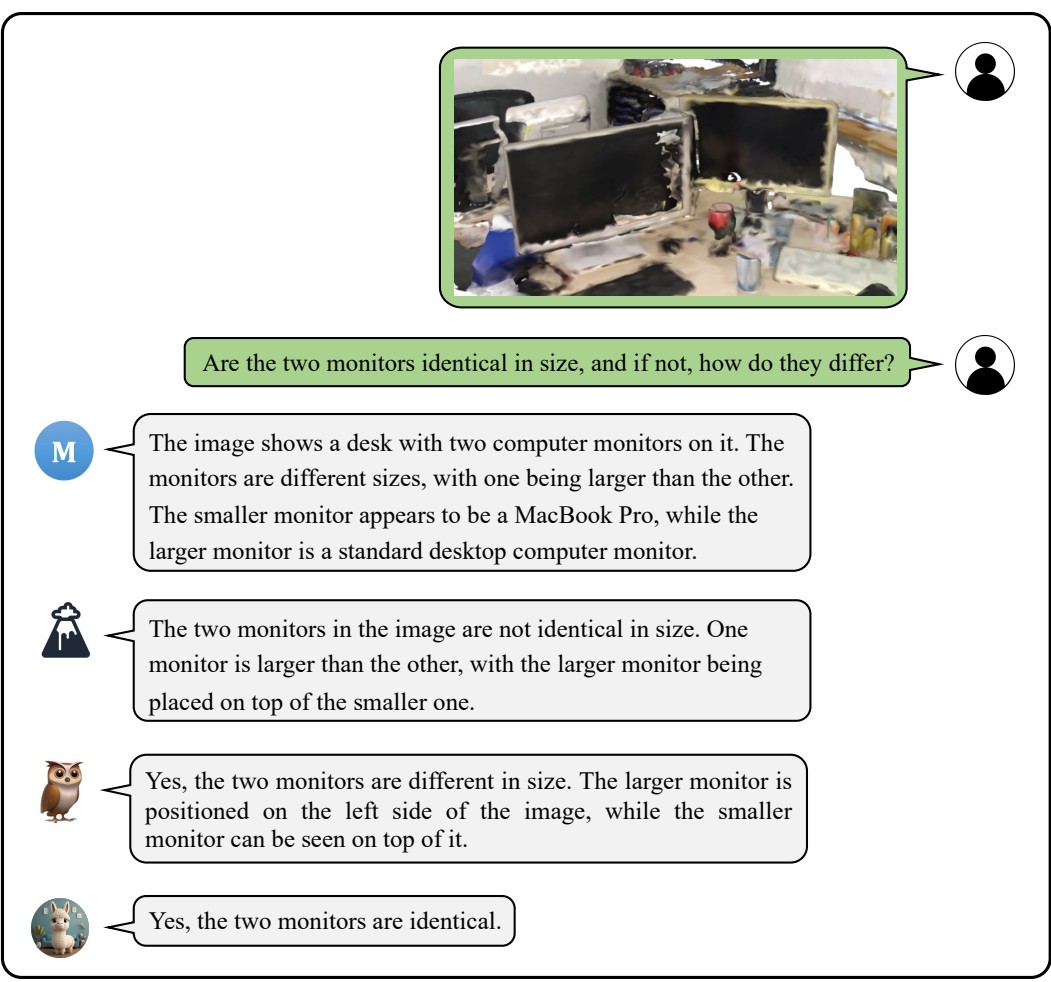

Figure 8: Example 2 of comparison between Chat-3D and 2D Multi-modal LLMs.

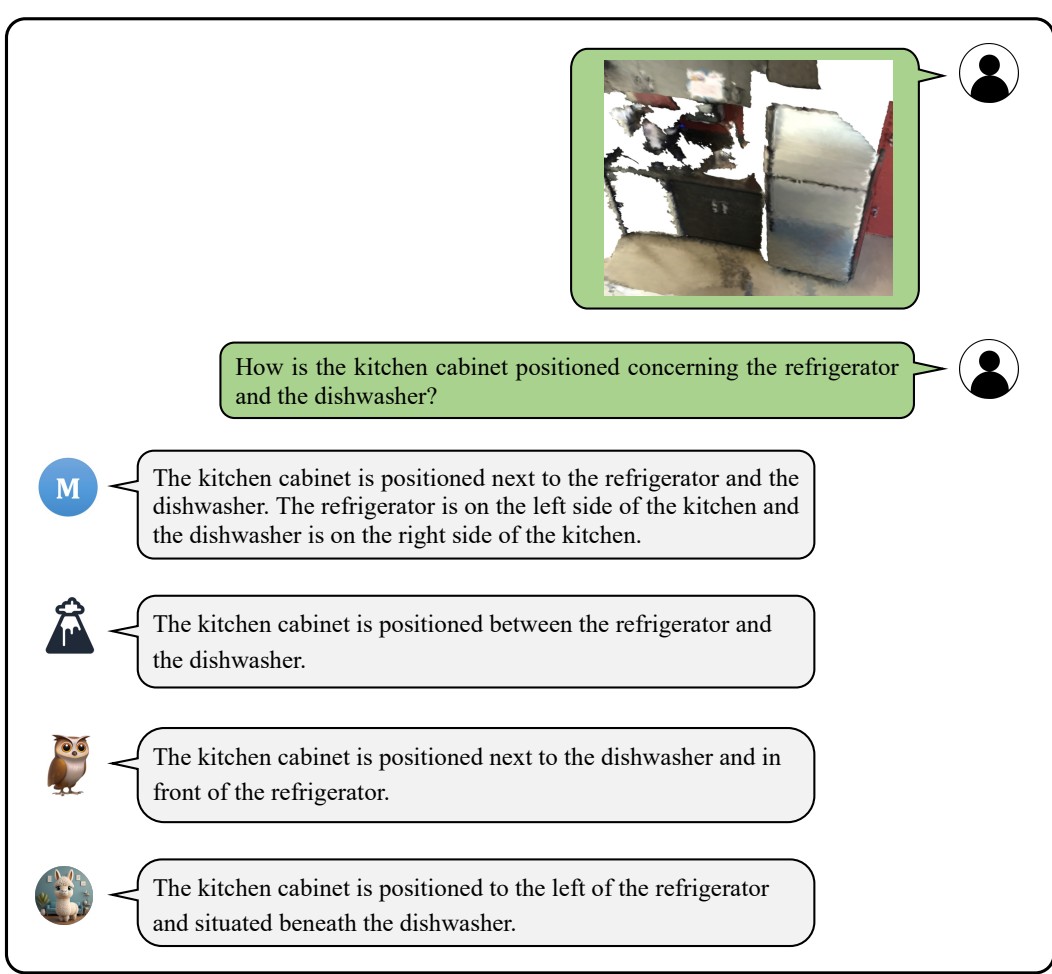

Figure 9: Example 3 of comparison between Chat-3D and 2D Multi-modal LLMs.

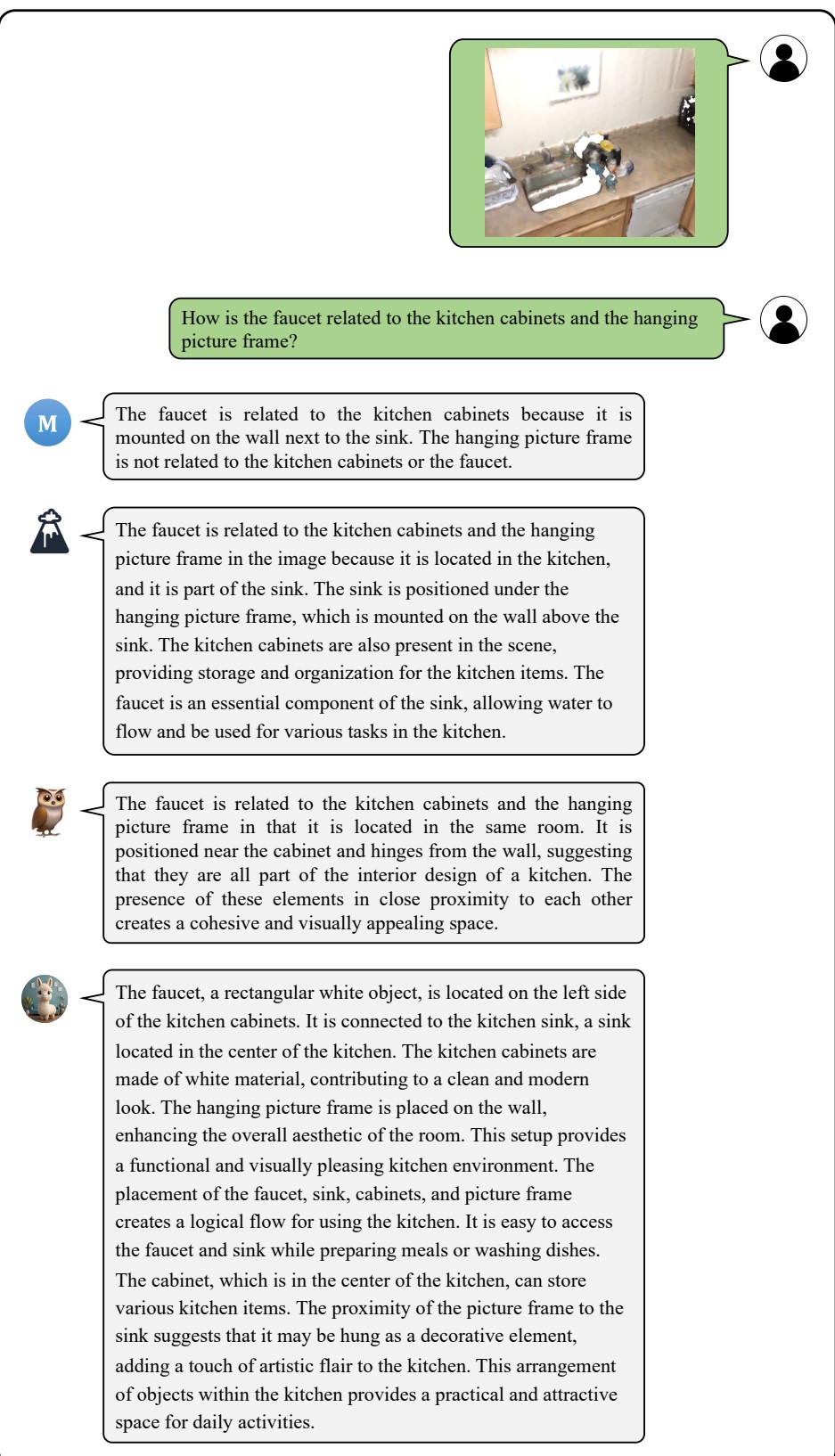

Figure 10: Example 4 of comparison between Chat-3D and 2D Multi-modal LLMs.

