# OpenReview forum: "Chat-3D: Data-efficiently Tuning Large Language Model for Universal Dialogue of 3D Scenes"
_ICLR.cc/2024/Conference — ICLR 2024 Conference Withdrawn Submission_

### Official Review · Reviewer_gi38 · 2023-10-27

**Soundness:** 3 good
**Presentation:** 2 fair
**Contribution:** 3 good
**Rating:** 3
**Confidence:** 4

**Summary:**

The paper presents Chat-3D, which is a multi-modal LLM to support dialogue of 3D scenes. Due to the limited availability of 3D-text data, it presents a novel three-stage training pipeline to tune the model more data-efficiently. Evaluation is done by GPT-4 and ScanQA dataset. It achieves 82.2% relative score compared with GPT-4, and comparable performance when compared with 3D-LLM on ScanQA dataset.

**Strengths:**

- The problem is both important and interesting. 3D-text pairs are data which are very hard to collect. If we can train a multi-modal LLM with 3D data efficiently, that can be a very influential paper.
- The proposed three-stage training makes use of both 3D object datasets and 3D scene datasets, which is novel. We typically see only one of them are used, or they are used together.

**Weaknesses:**

1. After reading the paper, I feel the primary contribution of Chat-3D is "data efficient". This is mentioned in the title as well as the conclusion of the primary benchmark ScanQA.

> This suggests that 3D-LLM heavily relies on the robust 2D VLM, which is pretrained on billion-level data. In contrast, Chat-3D solely utilizes 3D data for pretraining and fine-tuning, which is based on a much smaller data set. Nevertheless, it still manages to achieve competitive results compared to 3D-LLM (Flamingo), highlighting the effectiveness of our training architecture.

However, the paper fails to provide enough technical details to demonstrate the data efficiency.

- It is unclear what is Stage 1: 3D object alignment and Stage 2: 3D scene alighment are trained on. It mentions some general 3D object datasets such as ShapeNet and Objaverse. However, it does not mention what dataset it really uses. And more importantly, how the dataset is constructed? how many 3D-text pairs?
- It is not clear how the 3D instruction dataset is constructed (Stage 3). It does look like ScanRefer and ChatGPT are used. However, since this is a new dataset, I expect a detailed explanation of the dataset, including source of 3D models, and dataset statistics. This is also an important contribution of the paper, but very limited details are discussed in the paper.
- Without a detailed comparison of training data between Chat-3D and 3D-LLM, I am not convinced Chat-3D is data-efficient.

2. Right now the evaluation on object-centric dataset is an ablation study to show the effectiveness of three-stage tuning. However, I think actually it is possible to compare with external baselines (e.g. 3D-LLM, LLaVA). Then the scores will make more sense.

Some minor points:
- I do not fully understand if the relation module is necessary. It seems like a novel module proposed by the paper. However, I do not see ablation study on the module.

**Questions:**

In the rebuttal,

- Can you provide more details about the dataset (refer to weaknesses)?

In addition to that, I have some questions about the approach.

- Why $g$ and $f_e$ are separate feature encoders? I understand each object should have a unique bounding box, and location. But how do you obtain a unique color for each object? My intuition is RGB should be point cloud features.
- What does the location mean? Is it duplicate given that we have bounding boxes?

---

### Official Review · Reviewer_iHWf · 2023-10-31

**Soundness:** 2 fair
**Presentation:** 2 fair
**Contribution:** 2 fair
**Rating:** 3
**Confidence:** 4

**Summary:**

This paper present Chat-3D, a instruction-tuned 3DVLLM for 3D dialogue for 3D scenes. To train Chat-3D, the paper introduce a 3-stage training pipeline and construct a 3D instruction dataset. The author carry on experiments on visual QA datasets.

**Strengths:**

The research topic: 3D VLLM is a valuable research direction.

**Weaknesses:**

1. ScanQA experiments and metrics. First, the author should also present the EM in ScanQA evaluation to fairly compared with traditional discriminative baselines. Second, even on the BLEU score, which is more friendly to generative models,  Chat-3D cannot obtain obvious superiority compared to traditional discriminative model.

2. Lack of more dialogue analysis or experiments. The paper mainly present visual QA for reasoning and caption but claim a "universal dialogue". Human question can be very diverse. I think the author should provide more results to support this claim.

3. No open-world / open-vocabulary dialogue capability. When construct the instruction, the author just use the GT box in ScanRefer, which is limited to a small category space. Also, in the inference stage, the author mentioned that Chat-3D uses some close-set segmentation model or GT to segment objects, so how can Chat-3d do "universal dialogue" for potential category outside the close category set?

4. Fine-tune and inference pipeline should more clear.  The paper mentioned in method that it can use GT or 3D segmentor for 3D segmentation, and mentioned it use GT for training. However, I have not yet found which manner the paper used for ScanQA and the dialogue.

5. Lack of novelty. From the model training, dataset construction, I cannot exactly say what is unique contribution from. The dataset construction is similar to 3D-LLM and LLaVA.  The instruction-tuning is almost the same as LLaVA, but just replace the 2D vision model to 3D encoder. Even for the experiments and dialogue examples, they can only show some reasoning and caption capability of Chat-3D, and the results are not impressive. So, can the author has a indepth discuss on the novelty or contribution of Chat-3D?

**Questions:**

Please refer to the weakenss part.

---

### Official Review · Reviewer_9Qdi · 2023-11-01

**Soundness:** 2 fair
**Presentation:** 3 good
**Contribution:** 2 fair
**Rating:** 5
**Confidence:** 5

**Summary:**

This paper proposes Chat-3D, an LLM in 3D space for scene-level dialogue systems. To progressively align 3D with language embeddings, Chat-3D includes a three-stage training: object-level, inter-object, and instruction tuning. This work also collects a 3D instruction tuning dataset for future 3D-language research.

**Strengths:**

a) The proposed three-stage training is reasonable. The paper first conducts object-wise alignment and then inter-object relation learning. Afterwards, a 3D instruction dataset is adopted for chatting ability. Extensive ablation studies verify its effectiveness.

b) Chat-3D shows good conversation and reasoning performance for qualitative evaluation, outperforming 2D multi-modal LLMs, like LLaVA.

c) The training paradigm and dataset are meaningful to future 3D instruction tuning research.

**Weaknesses:**

a) The author somewhat overclaims its originality, since it is not the ***first*** attempt for using LLMs into 3D. The reviewer thinks '3D-LLM: Injecting the 3D World into Large Language Models' accepted by NeurIPS 2023 is the first work for 3D-scene LLM, which is released in July 2023.

b) This paper misses one discussion to related work. The author claims in Section 5.1 that "*We employ the pre-trained Point-Bind model to extract features for each object*". To my knowledge, *Point-Bind* itself also achieves a 3D LLM for object level released in June 2023, so this paper is required to discuss the relation with Point-Bind's 3D LLM.

c) The inference mode of Chat-3D must depend on a 3D segmenter and users' choice of objects. This probably means this method cannot generalize well to out-of-domain 3D scenes that the segmenter cannot handle.

d) The comparison to 3D-LLM is weak shown in Table 7. Is there any insight that how can Chat-3D improve the performance comparably to 3D-LLM?

**Questions:**

The reviewer is curious about the object-level QA ability of Chat-3D, since the stage-1 training is particularly for object-language alignment.

---

### Official Review · Reviewer_Qwp4 · 2023-11-02

**Soundness:** 2 fair
**Presentation:** 3 good
**Contribution:** 2 fair
**Rating:** 5
**Confidence:** 5

**Summary:**

This paper introduces Chat-3D, a dialogue system that merges the visual perceptual capabilities of 3D representations with Language Learning Models (LLMs) for 3D scene understanding.
Chat-3D aligns 3D features into LLMs, enabling them to perceive and interact with 3D environments. To address data scarcity, the authors devise a three-stage training strategy and present an object-centric 3D instruction dataset to improve reasoning and user interaction.
Experiments demonstrate Chat-3D's capabilities in universal dialogue and spatial reasoning based on 3D scenes.

**Strengths:**

1. The problem of introducing LLMs into 3D is new. Multi-modal LLM has gained significant progress in 2D scenes, which motivates us to unleash its potential in 3D space.

2. It's a good idea for relation module. Different from 2D images, 3D objects involve complicated inter-object relations as claimed in the paper and it's crucial to tackle its issue by specific relation designs.

**Weaknesses:**

1. A most related work, 3D-LLM, needs to be compared in Related Work with details. ImageBind-LLM-3D (also with Point-Bind backbone as Chat-3D) also needs to be discussed.

2. It seems only the foreground object features are fed into LLM. What if the background also include important information? It is also very cumbersome to adopt a 3D segmenter every time and manually select objects by users.

3. Can Chat-3D achieves 3D captioning of the entire scene or object navigation like 3D-LLM? The performance of grounding and QA compared to 3D-LLM is not impressive.

**Questions:**

See the weakness above